# Thermoplastic Starch with Maltodextrin: Preparation, Morphology, Rheology, and Mechanical Properties

**DOI:** 10.3390/ma17225474

**Published:** 2024-11-09

**Authors:** Lata Rana, Saffana Kouka, Veronika Gajdosova, Beata Strachota, Magdalena Konefał, Vaclav Pokorny, Ewa Pavlova, Zdenek Stary, Jaroslav Lukes, Marek Patocka, Veronika Hegrova, Ivan Fortelny, Miroslav Slouf

**Affiliations:** 1Institute of Macromolecular Chemistry of the Czech Academy of Sciences, Heyrovsky Sq. 2, 162 06 Prague, Czech Republic; lata2577rana@gmail.com (L.R.); kouka@imc.cas.cz (S.K.); gajdosova@imc.cas.cz (V.G.); beata@imc.cas.cz (B.S.); konefal.magdalenaa@gmail.com (M.K.); pokorny@imc.cas.cz (V.P.); pavlova@imc.cas.cz (E.P.); stary@imc.cas.cz (Z.S.); fortelny@imc.cas.cz (I.F.); 2Faculty of Mechanical Engineering, Czech Technical University in Prague, Technicka 4, 166 00 Prague, Czech Republic; jaroslav.lukes@fs.cvut.cz; 3NenoVision, Purkynova 127, 612 00 Brno, Czech Republicveronika.hegrova@nenovision.com (V.H.)

**Keywords:** thermoplastic starch, maltodextrin modification, lower viscosity, lower processing temperature

## Abstract

This work describes the preparation of highly homogeneous thermoplastic starches (TPS’s) with the addition of 0, 5, or 10 wt.% of maltodextrin (MD) and 0 or 3 wt.% of TiO_2_ nanoparticles. The TPS preparation was based on a two-step preparation protocol, which consisted in solution casting (SC) followed by melt mixing (MM). Rheology measurements at the typical starch processing temperature (120 °C) demonstrated that maltodextrin acted as a lubricating agent, which decreased the viscosity of the system. Consequently, the in situ measurement during the MM confirmed that the torque moments and real processing temperatures of all TPS/MD systems decreased in comparison with the pure TPS. The detailed characterization of morphology, thermomechanical properties, and local mechanical properties revealed that the viscosity decrease was accompanied by a slight decrease in the system homogeneity. The changes in the real processing temperatures might be quite moderate (ca 2–3 °C), but maltodextrin is a cheap and easy-to-add modifier, and the milder processing conditions are advantageous for both technical applications (energy savings) and biomedical applications (beneficial for temperature-sensitive additives, such as antibiotics).

## 1. Introduction

In recent years, starch-based polymer systems have demonstrated great potential as a sustainable alternative to petroleum-based plastics in many applications [1,2] due to their biodegradability [3,4] and biocompatibility [4,5]. Starches are abundant, readily available, and inexpensive, which makes them an attractive option for a wide range of *technical applications*, such as biodegradable packaging materials [6,7], transparent materials [8], textiles [9], and cosmetics [10]. As for *biomedical applications*, starch-based materials can be employed in the fields of drug delivery [11,12], antimicrobial and anti-cancer agents [13,14], biodegradable implants [15], or fluorescent nanoparticles for bioimaging [16,17].

Despite starch’s advantages and numerous applications, the processing of starch still shows some issues, limitations, and challenges. As the melting temperature of pure starch is substantially higher than its decomposition temperature, it is necessary to use low-molecular-weight plasticizers in order to prepare *thermoplastic starch* (TPS), which can be processed like standard thermoplastic polymers [4,18]. Many low-molecular-weight compounds such as glycerol, ethylene glycol, urea, and amino acids have been employed in TPS preparation, usually in combination with water [19,20,21]. Glycerol became the most widely used plasticizer due to its availability, biocompatibility, and the good mechanical performance of the final TPS [22]. Nevertheless, medium-molecular-weight compounds with similar structure (typically carbohydrates or analogous organic compounds with hydroxyl groups) were considered as well [23,24]. Sugars such as glucose, mannose, and fructose improved the water uptake and resulted in a more amorphous material [25], but they also reduced the tensile strength, glass transition temperature, and crystallinity [26]. Most of the low- and medium-molecular-weight plasticizers fail to remain bonded to starch molecules in the long term, leading to phase separation and/or retrogradation, accompanied by the deterioration of their mechanical properties [22,23,27].

Although many additives have been studied for plasticizing starch, there is limited information on preparing TPS using higher-molecular-weight compounds or oligomers as plasticizers and/or additives [28,29]. The comparison of TPS plasticized with diglycerol and polyglycerol has demonstrated that the polyglycerols yielded a more homogeneous TPS matrix [22]. Oligosaccharides, such as maltodextrins of different molecular weights, improved starch processability [30,31] and reduced starch recrystallization [32]. Of the higher-molecular-weight additives, maltodextrin is the most chemically compatible with TPS, as it is an oligomer obtained by partial hydrolysis of the extremely large starch molecules (Figure 1).

This study aims at the reliable preparation of homogeneous TPS at lower processing temperatures. Our starting point was a two-step preparation protocol which combines solution casting (SC) and melt mixing (MM) in order to yield highly homogeneous starch [21,33,34]. We added maltodextrin (MD) to starch powder during the first SC step in order to decrease the processing temperature during the second MM step. The main idea was that water and glycerol (Figure 1a) would act as a well-established combination of plasticizers yielding highly homogeneous TPS, while MD oligomer (Figure 1b) would act as a lubricant for the almost chemically identical starch polymer (Figure 1c). The MD was expected to decrease the average molecular weight of starch molecules and, as a result, the overall system viscosity and processing temperature. The typical processing temperature for TPS is around 120 °C. Its decrease would be beneficial for both technical applications (energy savings) and biomedical applications (in the case of temperature-sensitive additives, such as antibiotics [21,35]).

## 2. Materials and Methods

### 2.1. Materials

The wheat starch powder used in this study was delivered by Skrobarny Pelhrimov a.s. (Pelhrimov, Czech Republic). Maltodextrin made of starch A with a dextrose equivalent (DE) of 20 was supplied by Amylon (Ronov nad Sazavou, Czech Republic). Anhydrous glycerol (C_3_H_8_O_3_; >99%) and sodium bromide (NaBr; >99%) reagent grade were bought from Lach-Ner s.r.o. (Neratovice, Czech Republic). TiO_2_ anatase (average particle size ~20 nm) was supplied by Sigma-Aldrich (Sigma-Aldrich s.r.o., Praha, Czech Republic).

### 2.2. Preparation of Thermoplastic Starch

The six basic TPS systems prepared in this work are summarized in Table 1. All samples were prepared by a two-step preparation protocol, SC+MM, at a nominal processing temperature of 120 °C (6 samples; these samples were employed in the key experiments of the study, which are described in Section 3.2). Additionally, the maltodextrin-free samples (TPS and TPS/TiO_2_) were prepared by single-step MM and single-step SC protocols (4 samples; these samples are compared with the SC+MM samples in Section 3.1). The details of all three preparation protocols (SC, MM, and SC+MM) are described in our previous studies [33,34,36] and their brief descriptions follow (Section 2.2.1, Section 2.2.2 and Section 2.2.3). Furthermore, the TiO_2_-free samples (TPS, TPS/5MD, and TPS/10MD) were also prepared at two lower nominal processing temperatures of 100 °C and 90 °C (6 samples; these samples are characterized, discussed, and compared with the six basic samples prepared at 120 °C in Section 3.3).

#### 2.2.1. Single-Step Preparation by Solution Casting

The solution-cast TPS was prepared by premixing wheat starch, maltodextrin, glycerol, and water with a magnetic stirrer in a beaker for 30 min at room temperature. The premixed material was transferred to a stronger, mechanical stirrer RZR 2020 (Heidolph, Nuremberg, Germany) combined with a heating bath, where the temperature was elevated to 65 °C (temperature ramp ~5 °C/min). At the elevated temperature of 65 °C, the starch started to gelatinize, and then it was mixed for another 15 min while keeping the temperature at 65 ± 3 °C. The warm gelatinized starch was cast on a PE foil into ~2 mm thick film, and left to dry at ambient temperature for 3 days.

#### 2.2.2. Single-Step Preparation by Melt Mixing

The melt-mixed TPS was prepared using a twin-screw laboratory kneader (Brabender Plasti-Corder, Duisburg, Germany). Before the melt mixing, the starch powder (with maltodextrin) was premixed with glycerol (7/3 *v*/*v* ratio) in a common laboratory beaker (water was not used in the single-step MM as it would be unstable at the given processing conditions with temperatures > 100 °C). The mixture was put into the kneader chamber, which was pre-heated to the nominal temperature of 120 °C, and melt mixed for 8 min at 60 rpm rotation speed while recording the real processing temperature and torque moments. The mixed sample was compression molded to 2 mm thick plaque in two laboratory hot presses (Fontijne Grotnes, Vlaardingen, The Netherlands) using the following procedure: (i) 2 min in the first press at 130 °C and 50 kN, (ii) 2 min in the same press at 130 °C and 100 kN, (iii) 2 min in the second press at 100 °C and 100 kN, and (iv) the final slow cooling of the second press (still at 100 kN) with water to the ambient temperature (total cooling time ~15 min) in order to obtain the final TPS plaques.

#### 2.2.3. Two-Step Preparation by Solution Casting Followed by Melt Mixing

Two-step preparation procedure (SC+MM) combines SC (Section 2.2.1) and MM (Section 2.2.2) to achieve better homogeneity of the final TPS. The dried SC sample (57 g) was put into the laboratory mixer chamber (with volume 50 cm^3^) and processed as described above in Section 2.2.2.

### 2.3. Characterization of TPS Systems

#### 2.3.1. Light Microscopy

Light microscopy (LM) and polarized light microscopy (PLM) were employed in checking the homogeneity of TPS at lower magnifications. All measurements were performed in transmitted light, using a Nikon Eclipse 80i microscope (supplied by Laboratory Imaging, Praha, Czech Republic) equipped with a digital camera. The thin slices for LM and PLM (thickness 5 μm) were prepared using a rotary microtome (RM 2155; Leica, Wetzlar, Germany) and placed in a thin oil layer between the microscopic glasses.

#### 2.3.2. Scanning Electron Microscopy

A high-resolution scanning electron microscope (SEM; microscope MAIA3, Tescan, Brno, Czech Republic) was used to study the homogeneity of TPS at higher magnifications. The bulk TPS samples were fractured in liquid nitrogen, fixed in aluminum stub, and sputter-coated with a thin platinum layer (vacuum sputter coater SCD 050, Leica, Austria; thickness of the Pt layer: approx. 4 nm). The fracture surfaces were observed using secondary electron imaging (SEM/SE) and backscattered electron imaging (SEM/BSE) at an accelerating voltage of 3 kV.

#### 2.3.3. AFM-in-SEM Microscopy

A combined study of morphology and microscale properties was performed with a LiteScope (NenoVision, Brno, Czech Republic) Atomic Force Microscope (AFM) which is designed for integration into an SEM (so-called AFM-in-SEM). Integrating the AFM with an SEM enables precise navigation of the probe to the region of interest (e.g., interface between different phases of the TPS). The measurements were performed using a self-sensing AFM probe NenoProbe Conductive (NenoVision, Brno, Czech Republic). The probe stiffness was determined by the Sader method at 48 N/m. The measurements were made in the amplitude-modulated tapping regime (tip radius < 15 nm and resonance frequency 87 kHz). The device captured information on both the surface topography and local mechanical properties of the samples. The mechanical properties were examined through phase imaging, a technique that maps the phase difference between the driving signal applied to the AFM probe and the actual oscillations of the probe. This phase difference correlates with the sample’s mechanical characteristics, such as elasticity and adhesion. The SEM microscope utilized in all correlated AFM-in-SEM measurements was Versa SEM (Thermo Fisher Scientific, Brno, Czech Republic), operated in high-vacuum mode at an acceleration voltage of 0.5 kV.

#### 2.3.4. Wide-Angle X-Ray Scattering (WAXS)

The crystal structure variations in TPS systems were investigated through Wide-Angle X-ray Scattering (WAXS) analysis. The WAXS measurements were conducted using a pinhole camera (MolMet, Rigaku, Tokyo, Japan, modified by SAXSLAB/Xenocs) connected to a microfocused X-ray beam generator (MicroMax 003, Rigaku, Tokyo, Japan) operating at 50 kV and 0.6 mA (30 W). Scattering intensities were recorded using a vacuum-compatible hybrid photon-counting detector (Pilatus3 R 300 K, Pilatus, Basel, Switzerland), with a sample-to-detector distance of 50 mm and an exposure time of 4 h. The peak deconvolution procedure was performed using Fityk software version 1.3.1 [37]. The degree of crystallinity was determined by calculating the ratio of the crystalline peaks area to the total area under the WAXS curve.

#### 2.3.5. Rheology

Rheological properties of the TPS systems were measured in shear on a strain-controlled ARES G2 rheometer (TA Instruments, New Castle, DE, USA) using a parallel plate fixture with a diameter of 30 mm (plates with cross-hatched surface to prevent slipping). The thickness of the specimens was 2 mm. At first, the linear viscoelasticity region (LVER) was determined in view of the dependence of the storage modulus on the strain amplitude, which was measured at 120 °C at a frequency of 1 Hz. Next, the frequency sweep experiments were performed in a frequency range from 0.1 to 100 rad/s at a strain amplitude of 0.1% (always well within the LVER), and at a constant temperature of 120 °C. To ensure a uniform temperature in the specimen, all samples were equilibrated for 2 min prior to the start of each type of experiment. The frequency sweep was performed twice for each TPS specimen.

#### 2.3.6. Dynamic Mechanical Thermal Analysis

The thermomechanical properties of the TPS systems were measured in torsion by dynamic mechanical thermal analysis (DMTA) on specimens of rectangular platelet shape (40 mm × 10 mm × 2 mm), using an ARES G2 (TA Instruments, New Castle, DE, USA) in oscillatory mode, at a deformation frequency of 1 Hz. The deformation amplitude ranged from 0.01 to 3% (regulated automatically by the auto-strain function, in response to sample resistance). The investigated temperature range was from −90 to 120 °C, while the heating rate was 3 °C/min. The temperature dependences of the storage shear modulus (G’), of the loss modulus (G”), and of the loss factor tan(δ) were recorded.

#### 2.3.7. Nanoindentation

Nanoindentation mapping for indentation hardness and reduced elastic modulus was performed using a Hysitron TI950 TriboIndenter (Bruker, Billerica, MN, USA) with an automated method applied across the samples. The mapping pattern consisted of 10 × 10 indents with a 10 µm separation between indents. A displacement-controlled load function was employed, which included loading, holding, and unloading segments, each lasting 1 s, with a peak displacement of 1 µm. A lift height of 500 nm was executed before each indent to allow for correction of the zero-point contact during analysis. A Berkovich diamond tip was used as the indentation probe, and its calibrated tip area function was applied in the analysis of hardness and reduced elastic modulus upon the unloading segments.

## 3. Results and Discussion

In this section, the results are described and discussed in the order they were obtained and interpreted. In the first step (Section 3.1), we compared and re-verified the three possible preparation protocols of thermoplastic starch (TPS): solution casting (SC), melt mixing (MM), and their combination (SC+MM). In the second step (Section 3.2), we applied the most reliable protocol (SC+MM) to prepare TPS samples with 0, 5, and 10 wt.% of maltodextrin (MD) and characterized their properties via multiple methods (X-ray diffraction, rheometry, and thermomechanical analysis). In the third step (Section 3.3), we checked if the addition of MD could reduce the processing temperature while keeping the homogeneity of the systems (in terms of microstructure and micromechanical properties). In the last step (Section 3.4), the combination of all experimental results brought us to the conclusion that MD decreased the processing temperature of TPS at the slight expense of the overall system homogeneity in the microscale.

### 3.1. Optimal Preparation Protocol for TPS

In our previous study [33], we tested three preparation protocols for obtaining highly homogeneous thermoplastic starch (TPS). The first protocol was based on a single-step solution casting (SC), the second on a single-step melt mixing (MM) and the last was a two-step preparation combining both methods (SC+MM). In the first step of the current work, we re-verified our conclusion [33,34] that the combined SC+MM protocol yields TPS with the highest homogeneity. This is documented in Figure 2, which compares the three preparation protocols for TPS filled with 3 wt.% of TiO_2_ nanoparticles with an average size of ca 20 nm. The nanoparticles were added to the TPS just as a staining agent (morphological marker) that helped us visualize the morphology. Their impact on the structure, rheological, and thermomechanical properties was negligible, as documented in the following section. It is worth noting that the different types or batches of the initial starch powder can exhibit different properties, depending on the country of origin, starch type and/or manufacturer [21,36]. Nevertheless, the SC+MM protocol proved to be quite universal, reproducible, and applicable to all starch types studied in our laboratory.

Most researchers estimated the homogeneity of their TPS materials from SEM micrographs displaying the fracture surfaces of TPS [38,39,40]. These authors usually applied fracturing in liquid nitrogen (to minimize plastic deformations at the microscale [41]), the most common secondary electron imaging (SEM/SE mode, which yields topographic contrast [41]), and no markers (such as TiO_2_ nanoparticles employed here and in our previous studies [33,34]). However, the contrast between non-fully plasticized starch granules and the fully thermoplasticized matrix in SEM is quite weak, as illustrated in Figure 3. The non-plasticized granules in Figure 3 can be recognized mostly due to the presence of TiO_2_ nanoparticles that tend to envelope the non-plasticized granules, not penetrating inside. If the nanoparticles were not present (like in the great majority of the published TPS studies), then the SEM micrographs could bring us to the false conclusion that *all* TPS samples are reasonably homogeneous, regardless of the plasticization protocol. We conclude that many previous studies which tried to estimate the TPS homogeneity using SEM could have come to false positive results.

### 3.2. Impact of Maltodextrin on Structure and Properties of TPS

#### 3.2.1. Crystal Structure of TPS/MD Systems

Figure 4 displays WAXS results, which show the effects of processing and MD addition on the structure and crystallinity of TPS. For the source wheat starch powder, the WAXS data confirmed the A-type structure with characteristic strong reflections at 2θ = 15.3°, 2θ = 17.2° and 18.0° (doublet), and 2θ = 23.3° [42,43]. After thermomechanical processing (via MM, SC, or their combination marked as SC+MM), two types of crystallinity were observed. The first type was residual crystallinity associated with the incomplete disintegration of the original A-type wheat starch granules. The second one was the processing-induced crystallinity, commonly known as V-type crystallinity [33,42]. It occurs due to recrystallization of amylose in single helices during the cooling of the thermally processed material in the presence of plasticizers [27,44].

In all studied TPS samples, the V-type crystallinity was primarily represented by peaks at 2θ = 12.9°, 19.6°, and 22.0° [42,44,45]. An additional small peak at 5.3° in the low-angle region was observed. It was more pronounced for TPS prepared by single-step SC. This peak might be attributed to the longer-range order in crystalline structures (according to the general reciprocal relationship in diffraction, small diffraction angles in reciprocal/diffraction space correspond to large periodicities in real space).

As expected, the overall degree of crystallinity decreased after processing (SC and/or MM), which disrupted the original starch structure. The crystallinity decreased from ca 31% for native starch to 22% for samples processed by single-step SC and to ca 19% for the samples processed by our two-step SC+MM plasticization protocol. This corresponded to the PLM results (Figure 2), evidencing that the SC+MM processing resulted in a highly homogeneous TPS with disintegrated starch granules. The impact of MD and TiO_2_ addition on the overall crystallinity was negligible, within the experimental errors.

#### 3.2.2. Rheological Properties of TPS/MD Systems

Figure 5 compares the rheological properties of key TPS/MD systems at 120 °C: pure TPS after single-step MM (blue line), pure TPS after two-step SC+MM (orange), TPS with 3 wt.% of TiO_2_ nanoparticles after SC+MM (green), and TPS with 5 and 10 wt.% of maltodextrin, respectively (red and violet line, both samples after SC+MM). The data came from oscillatory shear rheometry (Section 2.3.4). The temperature of rheological measurements (120 °C) corresponded to the typical nominal temperature of starch processing during the melt mixing (Section 2.2).

The rheological properties of the first three samples without maltodextrin (TPS/0MD(MM), TPS/0MD(SC+MM) and TPS/0MD/3%TiO_2_(SC+MM)) were almost identical. This evidenced that the preparation protocol (MM vs. SC+MM) did not influence the final rheological behavior of the systems (compare the blue and orange line in Figure 5). The TiO_2_ nanoparticles, which were added just as a marker for the better morphology visualization (Section 3.1) did not change the final rheological properties either (compare the blue and green line in Figure 5).

The addition of maltodextrin made the TPS systems softer and less viscous. Both samples with MD (TPS/5MD(SC+MM) and TPS/10MD(SC+MM) exhibited lower values of all three rheological properties: *G*’ (a measure of the resistance to the elastic deformation), *G*” (a measure of the resistance to the viscous deformation), and |*η**| (a measure of the overall system viscosity). All three quantities decreased to ca 50% of their original values for the sample with 10 wt.% of MD (note the logarithmic scale in Figure 5). This indicated, in agreement with our assumptions, that the low-molecular-weight MD component could act as a lubricant, which could decrease the final TPS processing temperature.

The overall shape of all rheological curves confirmed that all the TPS systems behaved as solid polymers with very high molecular weight during the oscillatory shear tests. Firstly, we observed *G*’ > *G*” without a crossover in the whole frequency range (Figure 5a,b), which designated the solid, the *gel-like* behavior of all TPS systems (as opposed to *liquid-like* behavior if *G*” > *G*’). Secondly, the curves of *G*’ (Figure 5a) and |*η**| (Figure 5c) showed no plateau regions, which is the behavior typical either of polymers with very high molecular weight (such as starch or ultrahigh molecular weight polyethylene [46]), or crosslinked polymers (such as rubbers or epoxy resins [47]). Interestingly, the TPS samples behaved as solids during the oscillatory shear measurements at 120 °C (where they were subjected to *small deformations during oscillations*; Section 2.3.4), but they could be melt-mixed as viscous liquids at the same nominal temperature of 120 °C in the laboratory kneader (where they were subjected to *high deformations during rotation* of the screws in the chamber; Section 2.2).

#### 3.2.3. Thermomechanical Properties of TPS/MD Systems

Figure 6 summarizes the thermomechanical properties of key TPS systems in the temperature range from −90 °C to 120 °C. The samples and their abbreviations are the same as in the previous section, i.e., we compared the properties of TPS after MM (blue line in Figure 6), TPS after SC+MM (orange line), TPS with 3% of TiO_2_ after SC+MM (green line), and the two TPSs with 5 and 10 wt.% of maltodextrin (red and violet line, respectively). The thermomechanical properties of the non-modified TPS sample (TPS/0MD, blue line) were quite common, non-exceptional, and comparable to those obtained in previous studies [34,36].

In the low-temperature region up to 0 °C, the thermomechanical properties of all five TPS samples were quite similar. The TPS/10MD(SC+MM) sample (violet line in Figure 6) exhibited slightly higher values of both moduli (*G*’ and *G*”), which indicated possible complexities in the structure of TPS/MD samples. This pushed us, together with the DMTA results in the high-temperature region, to a more detailed study of possible morphological and micromechanical inhomogeneities in TPS/MD systems, which are discussed below in Section 3.3.

In the high-temperature region above 0 °C, the thermomechanical properties of the individual samples started to vary. Foremost, we could observe a clear difference between the samples without MD (green, orange, and blue lines) and the samples with the addition of MD (red and violet lines). Additionally, the MD-modified samples exhibited higher fluidity (higher *G*”/*G*’ ratio for *T* > 0 °C), which was manifested clearly in Figure 6c (we recall the definition of tan(δ) = *G*”/*G*’). Finally, the MD-modified samples showed three sharp peaks on the tan(δ) curve (at *T* ≈ −53, 15, and 65 °C), while the samples without maltodextrin exhibited just one sharp peak (at *T* ≈ −55 °C, just slightly shifted with respect to MD-modified samples) and two rather broad peaks at higher temperatures (at T ≈ 20 and 75 °C, markedly shifted with respect to MD-modified samples).

The DMTA tan(δ) curves of thermoplastic starches can exhibit as many as four different peaks [48,49,50]. Their presence, intensity, and position depend on the starch type, plasticization protocol, additives, fillers, crosslinking agents, etc. The peaks at lower temperatures are connected with multiple glass transition temperatures (*T*_g_), while the peak at the highest temperature is connected with the final melting of the plasticized material (*T*_m_) [24,48]. The first, low-*T*_g_ peak (usually between −60 and −20 °C) is associated with highly plasticized, amorphous regions where water and low-molecular-weight plasticizers, such as glycerol, are concentrated (these regions are often referred to as the “plasticizer-rich phase” [33,49,51]). The second, intermediate-*T*_g_ peak (usually between 0 and 30 °C) is attributed to less plasticized regions with a higher local concentration of starch chains (these regions are often referred to as “starch-rich phase” [49,52]). The third, high-*T*_g_ peak (usually between 30 and 100 °C) may occur if the *starch-rich phase* contains remnants of semicrystalline regions and/or amylose domains [48,53]). The last, *T*_m_ peak, usually between 120 and 150 °C, is connected with the final transition of TPS to the fully molten, viscous state [24,48]. In our case, we detected all three *T*_g_ peaks, but not the final *T*_m_ peak as the measurement was stopped at 120 °C. The split of the second and third peaks for MD-modified samples in comparison with samples without MD can be interpreted as a consequence of a stronger distinction between amorphous and crystalline domains in MD-modified samples. The decrease in the temperature of the two peaks with the increasing content of maltodextrin demonstrates its contribution to the mobility of the starch chains in the system.

### 3.3. Impact of Maltodextrin on Processing and Homogeneity of TPS

#### 3.3.1. Processing Temperatures and Torque Moments

Figure 7 illustrates how the addition of maltodextrin influenced the processing of TPS systems. The melt mixing step of our SC+MM preparation protocol was performed at three different nominal temperatures (*T*_n_), where *T*_n_ is the temperature to which the mixing chamber was pre-heated before the experiment.

The first *T*_n_ = 120 °C was the standard processing temperature during our starch plasticization protocol [33,34]. The other two temperatures (100 °C and 90 °C) were tested in order to verify if the addition of MD could make the systems more processable at milder conditions. We tested three samples with increasing concentrations of maltodextrin (TPS/0MD, TPS/5MD, and TPS/10MD). For the sake of clarity, the samples in Figure 7 are marked with analogous colors as in Figure 5 and Figure 6 (green for the pure TPS vs. red and violet for the TPS/MD systems).

The addition of maltodextrin decreased both the torque moments during melt mixing (TQ; Figure 7a) and the real processing temperatures (*T*_p_; Figure 7b). The decrease in TQ was a somewhat irregular function of the MD concentration as the TQ values were extremely sensitive to the exact loading of the chamber. The decrease in *T*_p_ with MD concentration was quite modest (from 2 to 3 °C), but it was very reliable and reproducible for all investigated nominal processing temperatures.

#### 3.3.2. Homogeneity of TPS Systems at Standard Processing Temperature

Figure 8 and Figure 9 document the impact of MD on the homogeneity and nanomechanical properties of TPS systems processed at *T*_n_ = 120 °C, respectively. The homogeneity of TPS (Figure 8) was visualized by three independent microscopic methods: LM of TPS/MD/TiO_2_ systems (Figure 8a–c), PLM of TPS/MD systems (Figure 8d–f), and SEM of TPS/MD/TiO_2_ systems (Figure 8g–i). As documented above (Figure 5 and Figure 6), the TiO_2_ nanoparticles were used just a morphological marker, whose influence on the rheological and thermomechanical properties of TPS systems was negligible.

The maltodextrin-free samples (Figure 8a–c; TPS/0MD samples with or without TiO_2_) prepared at the standard nominal processing temperature (*T*_n_ = 120 °C) were morphologically homogeneous. Their LM micrographs displayed a uniform dispersion of TiO_2_ nanoparticles (black in transmitted light), the PLM micrographs were almost featureless, and the SEM micrographs confirmed homogeneous dispersion of TiO_2_ at higher magnifications. In contrast, all samples with maltodextrin (Figure 8d–i; samples TPS/5MD and TPS/10MD with or without TiO_2_) exhibited lower homogeneity. Their LM micrographs displayed occasional agglomerates of swollen, non-fully homogenized starch granules (bright domains in LM micrographs, as the light-absorbing TiO_2_ nanoparticles did not penetrate inside the granules). The PLM micrographs showed two phases: the *morphologically smooth phase* like in the case of TPS/0MD and the *morphologically rough phase* corresponding to the agglomerates of the swollen granules (cf. Figure 8b with Figure 8e,h). Furthermore, the TPS/MD samples showed not only the swollen starch granules (the gray *rough phase* in PLM), but also a few original non-plasticized granules in both phases (the very bright spots in PLM). The few original granules appeared as the very bright spots in PLM due to their anisotropic nature, resulting in the high light depolarization. The swollen, but not fully homogenized starch granules appeared as light gray stripes in the *rough phase* due to their partially destroyed structure, leading to decreased anisotropy and a lower light depolarization. As the inhomogeneity of the TPS systems increased with the maltodextrin concentration, the TPS systems with more than 10 wt.% of MD were not studied in this work. The SEM micrographs of the TPS fracture surface are less suitable for the characterization of TPS homogeneity, as discussed above in Section 3.1, but careful comparison of TPS (Figure 8c) with TPS/MD (Figure 8f,i) reveals some swollen, non-fully homogenized starch granules also on the fracture surfaces of samples with maltodextrin. We conclude that the three independent microscopic techniques proved that MD addition resulted in a slight decrease in the starch homogeneity.

Figure 9 illustrates how the maltodextrin addition influenced local mechanical properties. The maltodextrin-free sample (Figure 9a; sample TPS/0MD) exhibited a smooth cut surface (Figure 9a, gray LM micrograph) and similar values of the elastic modulus, *E*_r_, in the whole investigated area (Figure 9a, color NHI map). The sample with maltodextrin (Figure 9b, sample TPS/5MD) showed both *smooth phase* and *rough phase* in the reflected light (Figure 9b; LM micrograph) and lower values of elastic moduli, *E*_r_, in the *rough phase* region (Figure 9b; NHI map, the lower left corner). Therefore, the *rough phase*, which was formed by the swollen, not-fully homogenized starch granules (see Figure 8 and the discussion in the previous paragraph), was somewhat softer than the fully homogenized matrix formed by the *smooth phase*. These nanoindentation results were consistent with the macroscale DMTA measurements, in which the TPS/MD samples at room temperature exhibited lower storage moduli, *G*’, in comparison with pure TPS (cf. Figure 6a and Figure 9b).

The different local mechanical properties of the smooth phase and rough phase were confirmed also by AFM-in-SEM measurements. In SEM imaging with secondary electrons, we could localize a rough region, and the AFM phase imaging indicated that the rough region was softer. The results are shown in Appendix A.

#### 3.3.3. Homogeneity of TPS Systems at Lower Processing Temperatures

Figure 10 documents how the overall homogeneity and transparency of the TPS/MD systems decreased with the maltodextrin concentration and nominal processing temperature, *T*_n_. All samples were photographed at the same transmitted light illumination produced by a light table (Figure 10a), saved as eight-bit grayscale images, and their overall transparency was evaluated by means of the morphological descriptor *MeanGray* (Figure 10b), which takes the value of 255 and 0 for a completely transparent and completely opaque sample, respectively. Both a higher concentration of MD and lower *T*_n_ lead to a decrease in the macroscale homogeneity and transparency. The *rough phase*, whose volume fraction was shown to increase with MD concentration (Figure 8), increased the macroscale opacity of the sample (Figure 10). Moreover, the results suggested that the fraction of *rough phase* increased with the decreasing nominal processing temperature.

Figure 11 illustrates that TPS samples plasticized at the lower nominal processing temperature (*T*_n_ = 100 °C) contained a higher amount of the *rough phase* than the samples processed at the standard starch processing temperature (*T*_n_ = 120 °C). A high volume fraction of the *rough phase* was found even in the sample without maltodextrin (Figure 11a). For the lowest processing temperature (*T*_n_ = 90 °C), the amount of the rough phase was even higher. In conclusion, Figure 10 and Figure 11 confirmed that lower temperatures resulted in the lower TPS homogeneity that was associated with the higher content of the *rough phase*. Our preliminary results suggested that the combination of slightly increased temperature during SC (*T* ≈ 70 °C) combined with an intermediate nominal processing temperature during MM (*T*_n_ = 110 °C) might be optimal for obtaining highly homogeneous TPS/MD systems. This is a subject of our ongoing research.

### 3.4. Benefits and Drawbacks of TPS/MD Systems

#### 3.4.1. Lower Processing Temperature of TPS/MD Systems

The main advantage of TPS/MD systems consisted in the decrease in the real processing temperature with respect to pure TPS (Figure 7). The decrease was quite modest (around 2–3 °C), but it was very reproducible and showed a logical correlation with torque moments (compare Figure 7a,b). Even such a moderate decrease could bring real practical benefits. Maltodextrin is cheap and its addition to starch during plasticization is straightforward, not requiring a substantial modification of existing protocols. The decrease in the real processing temperature should enable us to decrease the nominal processing temperature (i.e., the pre-heating of the mixing chamber) while keeping a similar structure and properties of the final material. For large-scale *technical applications*, this could result in fair energy savings [55,56]. For selected *biomedical applications*, such as the addition of antibiotics to TPS in order to obtain a biodegradable system with a local drug release, even a small decrease in the processing temperature might help in maintaining the high antibiotic activity [21,34].

#### 3.4.2. Lower Homogeneity and Stiffness of TPS/MD Systems

All thermoplastic starches are soft multiphase materials with complex morphology as documented by many previous studies and re-confirmed here by LM, SEM, WAXS, and DMTA results (Section 3.1 and Section 3.2 and references therein). The maltodextrin-free TPS is morphologically homogeneous only if we use the optimized plasticization protocol (compare Figure 2a,b with Figure 2c). Moreover, even the morphologically homogeneous TPSs consist of several phases, as evidenced by multiple glass transitions detected by DMTA (Section 3.2.3 and discussion therein). This study has demonstrated that MD acted as an efficient, cheap, and compatible lubricant for TPS, decreasing the viscosity (Figure 5c) and real processing temperatures (Figure 7b). However, the lower viscosity of the TPS/MD matrix resulted in a lower overall stiffness at temperatures above 0 °C (Figure 5a,b) and slightly less homogeneous material (Figure 8 and Figure 9). Furthermore, the TPS/MD inhomogeneity tended to increase with increasing MD concentration at all processing temperatures (Figure 10 and Figure 11). For typical *technical applications* of TPS (agriculture, packaging, etc.) the slightly lower overall stiffness and microscale inhomogeneities indicated by nanoindentation might be of minor importance, while the lower processing temperature is a clear advantage. For selected *biomedical applications* (such as biodegradable implants with controlled drug release), the highest possible homogeneity could be an issue. Nevertheless, both the available literature [57,58] and our preliminary results (mentioned at the end of Section 3.3.3) suggest that the lower-viscosity TPS matrix can be plasticized better when we slightly increase the temperature during the SC. The further optimization of the plasticization protocol of TPS/MD systems, followed by microbiological testing of TPS/MD/antibiotics systems, will be a logical continuation of this work.

## 4. Conclusions

We have prepared highly homogeneous thermoplastic starches (TPSs) with the addition of 0, 5, and 10 wt.% of maltodextrin (MD) and 0 and 3 wt.% of TiO_2_ nanoparticles. The preparation of the highly homogeneous TPSs was based on our two-step preparation protocol, which comprised solution casting (SC) followed by melt mixing (MM). All prepared systems were thoroughly characterized by multiple techniques, such as LM, SEM, rheological measurements, DMTA, in situ measurement of processing temperature and torque moments during melt mixing, and mapping of local mechanical properties by nanoindentation. The main results can be summarized as follows:Maltodextrin acted as a lubricating agent that decreased the viscosity, torque moments, and processing temperatures of TPS/MD systems in comparison with the control, maltodextrin-free samples.The desired decrease in TPS/MD processing temperatures was accompanied by a slight decrease in system homogeneity, which was not critical at the macroscale, but it could be detected by multiple microscale techniques, such as LM, PLM, and nanoindentation.Further research will be focused on (i) further optimization of the preparation protocol in order to achieve lower processing temperatures while keeping the maximal homogeneity of the prepared material, and (ii) evaluation of the effect of MD addition on the properties of real systems, such as TPS/MD blends for technical applications and TPS/MD-based biodegradable systems for the local release of antibiotics.

## Figures and Tables

**Figure 1 materials-17-05474-f001:**
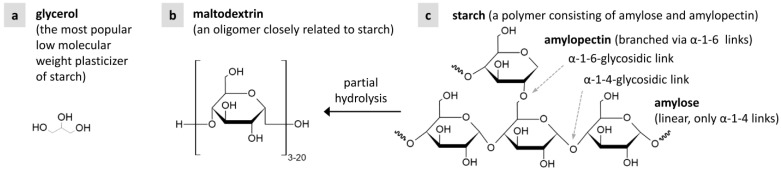
Chemical structures of (**a**) glycerol, (**b**) maltodextrin, and (**c**) starch.

**Figure 2 materials-17-05474-f002:**
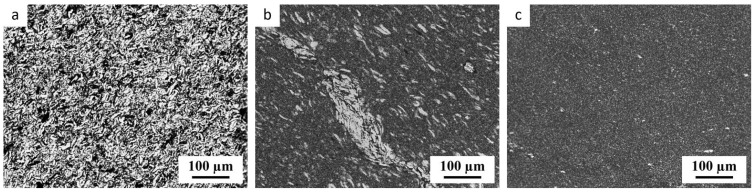
LM micrographs showing thin sections of TPS with 3 wt.% of TiO_2_ nanoparticles. The TPS was prepared in three different ways: (**a**) the single-step solution casting (SC), (**b**) the single-step melt mixing (MM), and (**c**) the two-step protocol including SC followed by MM (SC+MM). The samples were observed in transmitted light, in which TiO_2_ nanoparticles appear dark due to light absorption. The non-fully-plasticized starch granules appeared bright, as the dark TiO_2_ nanoparticles could not penetrate inside them.

**Figure 3 materials-17-05474-f003:**
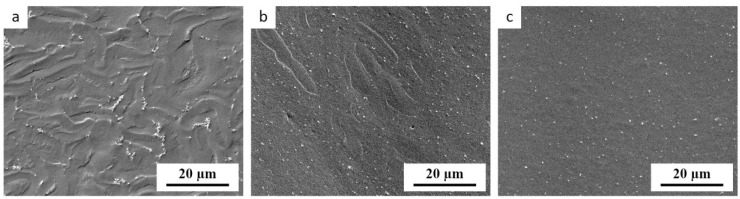
SEM micrographs showing fracture surfaces of TPS with 3 wt.% of TiO_2_ nanoparticles. The TPS was prepared in three different ways: (**a**) the single-step SC, (**b**) the single-step MM, and (**c**) the two-step protocol (SC+MM). The samples were observed in the secondary electron mode at the accelerating voltage of 3 keV, in which the TiO_2_ nanoparticles (or their small agglomerates) appear as small bright spots due to topographic contrast. The non-fully plasticized starch granules can be revealed by the fact that the bright TiO_2_ agglomerates tend to envelop them.

**Figure 4 materials-17-05474-f004:**
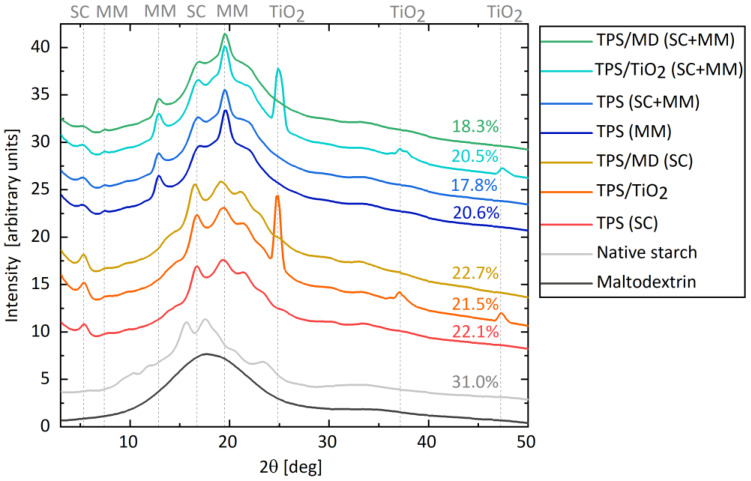
WAXS diffraction patterns and degrees of crystallinity of the initial starch and selected TPS systems. From bottom to top: the source maltodextrin powder; the source wheat starch powder; three TPS systems plasticized by the single-step SC protocol (pure TPS: TPS(SC); TPS with 3 wt.% of TiO_2_: TPS/TiO_2_(SC); TPS with 10 wt.% of MD: TPS/MD(SC)); one TPS system plasticized by the single-step MM protocol; and the final three TPS systems plasticized by the two step SC+MM protocol ((pure TPS: TPS(SC+MM); TPS with 3 wt.% of TiO_2_: TPS/TiO_2_(SC+MM); TPS with 10 wt.% of MD: TPS/MD(SC+MM)). The strong peaks typical of MM, SC, and TiO_2_ are marked by vertical dashed lines.

**Figure 5 materials-17-05474-f005:**
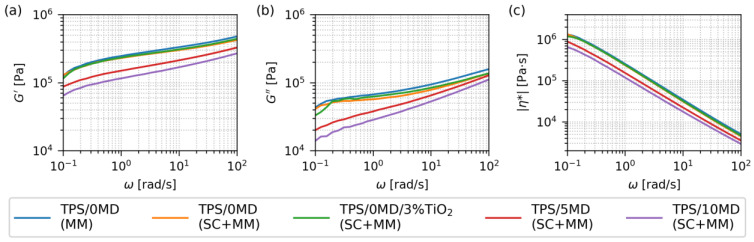
Rheological properties of selected TPS systems, measured by oscillatory shear rheometry at 120 °C: (**a**) storage modulus G’, (**b**) loss modulus G”, and (**c**) absolute values of complex viscosity |η*|. The measured samples comprised pure TPS after melt mixing (TPS/0MD(MM); blue line), pure TPS after melt mixing and solution casting (TPS/0MD(SC+MM); orange line), pure TPS after melt mixing and solution casting with 3 wt.% of TiO_2_ nanoparticles (TPS/0MD/3%TiO_2_(SC+MM); green line), and two samples with 5 and 10 wt.% of maltodextrin (TPS/5MD(SC+MM) and TPS/10MD(SC+MM), marked with red and violet lines, respectively). Note that *G*’ and *G*” are plotted with the same *y*-axis limits in order to facilitate direct comparison of the two quantities.

**Figure 6 materials-17-05474-f006:**
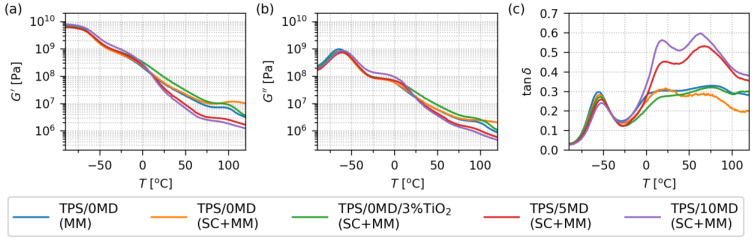
Thermomechanical properties of the selected TPS systems, measured by DMTA in rectangular torsion mode in temperature range −90–120 °C: (**a**) storage modulus G’, (**b**) loss modulus G”, and (**c**) damping factor tan(δ). The measured samples were the same as in the case of rheological analysis: TPS after melt mixing (TPS/0MD(MM); blue line), pure TPS after melt mixing and solution casting (TPS/0MD(SC+MM); orange line), pure TPS after melt mixing and solution casting with 3 wt.% of TiO_2_ nanoparticles (TPS/0MD/3%TiO_2_(SC+MM); green line), and two samples with 5 and 10 wt.% of maltodextrin (TPS/5MD(SC+MM) and TPS/10MD(SC+MM), marked with red and violet lines, respectively). Note that *G*’ and *G*” are plotted with the same *y*-axis limits in order to facilitate the direct comparison of the two quantities.

**Figure 7 materials-17-05474-f007:**
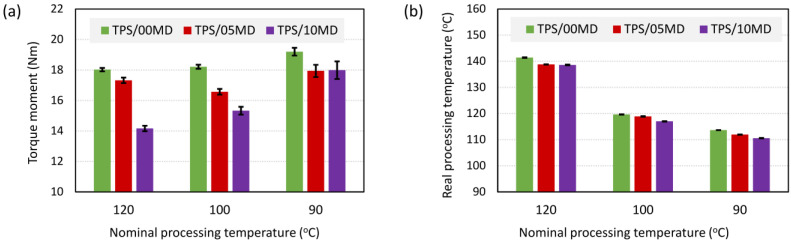
Processing parameters during the melt mixing of TPS/MD systems: (**a**) torque moments and (**b**) real processing temperatures. The investigated TPS/MD systems contained 0, 5, and 10 wt.% of maltodextrin and were denoted as TPS/00MD, TPS/05MD, and TPS/10MD, respectively. The melt mixing chamber was pre-heated to the selected nominal processing temperature (120, 100, or 90 °C) and the processing parameters were measured in situ during the melt mixing process. The values in both plots represent averaged steady-state values in the second half of the melt mixing; the error bars represent standard deviations (in (**b**) the standard deviations are quite small).

**Figure 8 materials-17-05474-f008:**
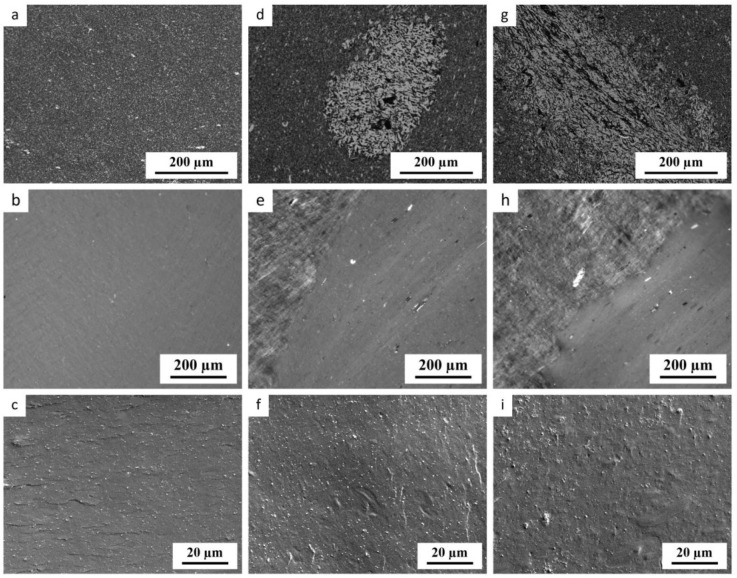
Homogeneity of TPS/MD systems processed at *T*_n_ = 120 °C as observed by LM, PLM, and SEM. Columns from left to right: (**a**–**c**) pure TPS, (**d**–**f**) TPS with 5 wt.% of MD, and (**g**–**i**) TPS with 10 wt.% of MD. Rows from top to bottom: (**a**,**d**,**g**) LM showing the thin sections of the TPS/MD systems with 3 wt.% of TiO_2_ nanoparticles, (**b**,**e**,**h**) PLM showing the thin sections of the TPS/MD systems without TiO_2_ nanoparticles, and (**c**,**f**,**i**) SEM/SE micrographs showing the fracture surfaces of TPS/MD systems with 3 wt.% of TiO_2_ nanoparticles.

**Figure 9 materials-17-05474-f009:**
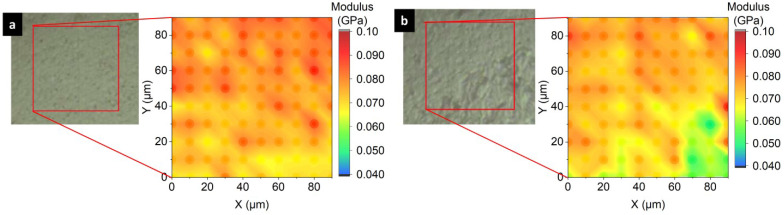
Homogeneity of TPS/MD systems processed at *T*_n_ = 120 °C as observed by NHI mapping: (**a**) TPS/00MD and (**b**) TPS/5MD sample. The reflected light micrographs (gray images) show the regions selected by means of the light microscope within the nanoindenter for the nanomechanical analysis. The maps of nanomechanical properties (color images) show the values of reduced elastic modulus (*E*_r_) calculated for each of the 10 × 10 indents taken in the square grid with a step of 10 μm.

**Figure 10 materials-17-05474-f010:**
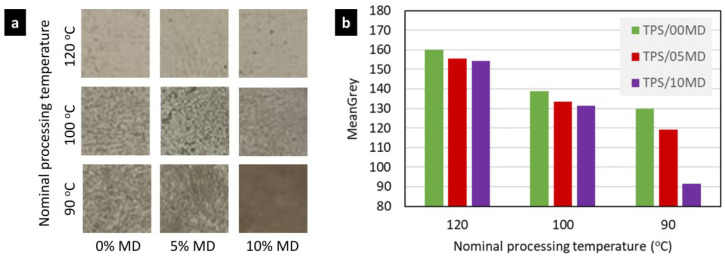
Comparison of the homogeneity of TPS/MD systems processed at temperatures 120 °C, 100 °C, and 90 °C: (**a**) macrophotographs of 2 mm thick plates in homogeneous transmitted light and (**b**) image analysis of the macrophotographs, showing the decrease in transparency (using the morphological descriptor *MeanGrey*, which ranges from 255 for 100% transparent samples to 0 for completely opaque samples). The image analysis was performed with ImageJ version 1.53e [54].

**Figure 11 materials-17-05474-f011:**
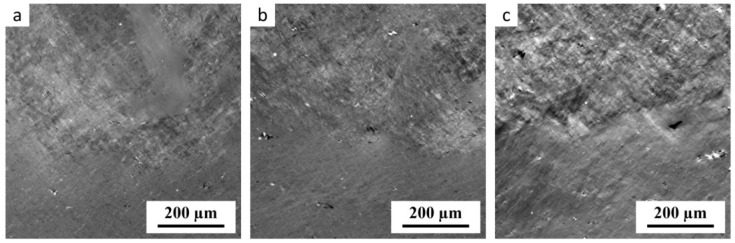
PLM inhomogeneous structures in TPS/MD systems, which were melt mixed at the lower nominal processing temperature *T*_n_ = 100 °C: (**a**) TPS/0MD, (**b**) TPS/5MD, and (**c**) TPS/10MD. In all three micrographs, the *rough phase* is at the top and the *smooth phase* is at the bottom. The micrographs were obtained in the same way as those in Figure 8d–f.

**Table 1 materials-17-05474-t001:** List of prepared TPS/MD/TiO_2_ solid systems.

Sample ID	Sample Composition *	Starch/Glycerol (*v*/*v* Ratio) **	Starch/Water (*w*/*w* Ratio) ***
TPS/0MD	TPS	7/3	1/6
TPS/TiO_2_	TPS/TiO_2_ (3%)	7/3	1/6
TPS/5MD	TPS/MD (5%)	7/3	1/6
TPS/5MD/TiO_2_	TPS/MD (5%)/TiO_2_ (3%)	7/3	1/6
TPS/10MD	TPS/MD (10%)	7/3	1/6
TPS/10MD/TiO_2_	TPS/MD (10%)/TiO_2_ (3%)	7/3	1/6

* Both TiO_2_ and MD are given in wt.% with respect to the initial starch powder. ** The volume ratio of initial starch powder to glycerol was the same for all samples and preparation protocols. *** The mass ratio of the initial starch powder to water was relevant only to SC and SC+MM samples; in the SC preparation, most of the water evaporated during the 3-day drying period of the solution-cast samples.

## Data Availability

The original contributions presented in the study are included in the article, further inquiries can be directed to the corresponding author.

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
