# Peer review of "Thermoplastic Starch with Maltodextrin: Preparation, Morphology, Rheology, and Mechanical Properties"

_materials, 2024, doi:10.3390/ma17225474_

Round 1
Reviewer 1 Report
Comments and Suggestions for Authors
1. Please check the whole manuscript and revise the grammar errors to make it more readable.
2. Please revise the references styles to follow the uniform format. Moreover, more recent work since 2023 should be cited accordingly.
3. Please compare the samples with similar materials from recent work to show their advantages.
4. More details for the characterization should be added,
Author Response
Comment 1: Please check the whole manuscript and revise the grammar errors to make it more readable.
Response 1: We did our best to re-check the whole manuscript and to correct the remaining grammar errors (these minor changes are not highlighted in the manuscript).
Comment 2: Please revise the references styles to follow the uniform format. Moreover, more recent work since 2023 should be cited accordingly.
Response 2: In fact, most of the literature cited in the original manuscript has been quite recent. From our 58 references, almost 20 were published at 2020 or later, and more than 20 were published at 2010 or later. Nevertheless, we re-checked the latest literature as the reviewer suggested and added two very recent studies to the first paragraph of the Introduction (Agarwal et al. 2023; Shi et al. 2025). All references in the revised manuscript use the MDPI format as defined in Zotero.
Comment 3: Please compare the samples with similar materials from recent work to show their advantages.
Response 3: The thermomechanical properties of the control, non-modified TPS sample were quite common, non-exceptional, and comparable to those obtained in the previous studies (this statement with suitable references was added at the end of the first paragraph of section 3.2.3). The mechanical properties of maltodextrin-modified starches were not much different (as described in the rest of section 3.2.3). The homogeneity of our starches was higher than in most previous studies (as discussed in detail in section 3.1).
Comment 4: More details for the characterization should be added.
Response 4: We re-checked the manuscript carefully for possible missing details concerning the characterization methods. A few more details were added to the Experimental part. They are all marked with the red font. The biggest change was the complete re-writting of section 2.2 in order to clarify which samples were characterize by which methods.
Reviewer 2 Report
Comments and Suggestions for Authors
Thermoplastic starch (TPS) with different maltodextrin (MD) contents and with 0 or 3% by weight of TiO2 nanoparticles added was developed. Rheological studies conducted under typical starch processing conditions showed that maltodextrin acted as a lubricant, reducing the viscosity of the system. As a result, in situ measurements during mixing confirmed that the torques and actual processing temperatures of all TPS/MD systems were lower compared to pure TPS. It was noted that maltodextrin is a cheap and easy-to-apply modifier, and milder processing conditions are beneficial for both technical and biomedical applications. The work is interesting, although I have a few remarks and comments.
1. The abstract should be prepared in an impersonal form
2. Why was the addition of 5% and 10% maltodextrin chosen, and not other contents?
3. Why was the share of 3% TiO2 chosen, and not other contents?
4. How was the homogeneity/homogenization of the mixing of the composition components assessed?
5. Was the Rabinowitsh and Bagley correction taken into account when assessing the rheology?
6. For how many samples were the tests performed?
7. Was statistical analysis performed?
8. What method of experiment planning was used?
9. The literature introduction should be expanded with the cited content - sometimes even 4 literature items are cited for one piece of information.
10. I am missing error bars on the graphs, e.g. for Figure 7.
11. Figure 10 a is hardly visible.
Author Response
Introductory comment: Thermoplastic starch (TPS) with different maltodextrin (MD) contents and with 0 or 3% by weight of TiO2 nanoparticles added was developed. Rheological studies conducted under typical starch processing conditions showed that maltodextrin acted as a lubricant, reducing the viscosity of the system. As a result, in situ measurements during mixing confirmed that the torques and actual processing temperatures of all TPS/MD systems were lower compared to pure TPS. It was noted that maltodextrin is a cheap and easy-to-apply modifier, and milder processing conditions are beneficial for both technical and biomedical applications. The work is interesting, although I have a few remarks and comments.
Response to introductory comment: We than the reviewer for their overall positive evaluation. The comments below are answered point-by-point.
Comment 1: The abstract should be prepared in an impersonal form
Response 1: We used the personal form only in the very first sentence of the abstract. This was changed as the reviewer suggested. The modification is marked with the red font.
Comment 2: Why was the addition of 5% and 10% maltodextrin chosen, and not other contents?
Response 2: Higher concentrations of maltodextrin decreased the overall homogeneity of the system, as discussed in detail in section 3.3. We slightly modified the text in the second paragraph of section 3.3.2 so that it was clearer. The change is marked with the red font in the revised manuscript.
Comment 3: Why was the share of 3% TiO2 chosen, and not other contents?
Response 3: TiO2 nanoparticles were added just as a morphological marker (or, in other words, as a staining agent), which helped us to visualize and evaluate the overall homogeneity of the TPS systems. This is discussed at the very beginning of the Results section (section 3.1.). The key sentence was slightly modified and marked with the red font in the revised manuscript.
Comment 4: How was the homogeneity/homogenization of the mixing of the composition components assessed?
Response 4: The homogeneity of the samples was assessed by means of light microscopy (LM), polarized light microscopy (PLM) and scanning electron microscopy (SEM) as described in the Result section (namely in subsections 3.1. and 3.3).
Comment 5: Was the Rabinowitsh and Bagley correction taken into account when assessing the rheology?
Response 5: Rabinowitsh and Bagley corrections are applied in capillary rheometry to obtain values of true share rate and true shear stress, respectively. However, our tests were performed as oscillatory shear measurements (as described in section 2.3.6), and hence the Rabinowitsch and Bagley corrections were not applied, as the rheometer software processed the oscillatory data to obtain the components of the complex modulus, rather than shear rate or shear stress. The viscosity was calculated from the moduli in an additional step. The necessary corrections for our type of measurement were managed automatically by the instrument’s analysis software.
Comment 6: For how many samples were the tests performed?
Response 6: We prepared six different TPS/MD systems, as specified in Table 1. All samples were prepared at nominal processing temperature of 120C using two-step preparation protocol SC+MM (6 samples). Additionally, the maltodextrin-free samples (TPS and TPS/TiO2) were prepared by single-step MM and single-step SC protocol (4 samples). Moreover, the TiO2-free samples (TPS, TPS/5MD and TPS/10MD) were prepared also at two lower nominal processing temperatures of 100C and 90C (6 samples). Therefore, we prepared 6 basic samples, but when we consider also various preparation protocols, the total number of samples was 16 (6 + 4 + 6, as described above). Section 2.2. was re-written completely so that this was clearer.
Comment 7: Was statistical analysis performed?
Response 7: No specific statistical analysis was performed (except for common calculations of statistical means and standard deviations).
Comment 8: What method of experiment planning was used?
Response 8: We have not used any specific method of experiment planning. Nevertheless, the brief description of how the research was conducted is given at the very beginning of section 3 (the first, introductory paragraph in the Result section).
Comment 9: The literature introduction should be expanded with the cited content - sometimes even 4 literature items are cited for one piece of information.
Response 9: The references in the first paragraph were arranged in smaller groups as the reviewer suggested. Moreover, we improved the description of the references, removed two less interesting references that just re-confirmed the same piece of information, performed additional search in the literature and added two very recent and more relevant references (Agarwal et al. 2023; Shi et al. 2025).
Comment 10: I am missing error bars on the graphs, e.g. for Figure 7.
Response 10: The error bars were added to Figure 7 (they represent the standard deviations, which were calculated from the same data as the average values, i.e. from the steady-state values in the second half of the melt mixing). The legend of Fig. 7 was updated accordingly.
Comment 11: Figure 10a is hardly visible.
Response 11: We suppose that the reviewer means that the Figure 10a has low resolution. This is caused by the fact that the evaluation of macroscale homogeneity was based on macrophotographs of the original 2 mm thick plates (i.e. not LM micrographs of 5 micrometer thin sections, which exhibit higher resolution). The reviewer is right that the macrophotographs are not as sharp as the micrographs in all other images of the submitted manuscript, but we believe they are better for this specific purpose because of two reasons: (i) The macrophotograph show bigger part of the sample than the light micrographs – this makes the results less sensitive to local fluctuations and more reliable. (ii) All macrophotographs could be recorded together, at exactly the same illumination with a digital camera when all samples were placed on a light table next to each other – to keep exactly the same illumination in the light microscope, where the micrographs have to be recorded one-by-one, is not so easy and consistent.
Round 2
Reviewer 2 Report
Comments and Suggestions for Authors
The authors of the publication took my comments into account. I recommend this article for publication.